# External Ear Canal Evaluation in Dogs with Chronic Suppurative Otitis Externa: Comparison of Direct Cytology, Bacterial Culture and 16S Amplicon Profiling

**DOI:** 10.3390/vetsci9070366

**Published:** 2022-07-18

**Authors:** Caroline Leonard, Damien Thiry, Bernard Taminiau, Georges Daube, Jacques Fontaine

**Affiliations:** 1Department for Clinical Sciences, Faculté de Médecine Vétérinaire, University of Liege, 4000 Liege, Belgium; jacques.fontaine@uliege.be; 2Bacteriology Laboratory, Department of Infectious and Parasitic Diseases, FARAH and Faculty of Veterinary Medicine, University of Liege, 4000 Liege, Belgium; damien.thiry@uliege.be; 3Fundamental and Applied Research for Animal & Health (FARAH), Food Science Department, Faculty of Veterinary Medicine B43b, University of Liege, 4000 Liege, Belgium; bernard.taminiau@uliege.be (B.T.); georges.daube@uliege.be (G.D.)

**Keywords:** suppurative otitis externa, 16S amplicon profiling, bacterial culture, bacterial susceptibility, canine

## Abstract

**Simple Summary:**

Discrepancy between the results of cytology and bacterial culture methods is sometimes observed in canine otitis externa. The objective of this study was to compare different techniques: direct cytology, aerobic bacterial culture and 16S amplicon profiling. Samples from twenty ears with chronic suppurative otitis externa were analysed. Good correlation between cytology and bacterial culture was observed in 60% of samples. Good correlation between bacterial culture and 16S amplicon profiling was observed in only 10% of samples when the overall 16S amplicon profiling results were used. Nevertheless, the correlation improved to 70% when bacterial species with a relative abundance >10% (considered as insignificant) were taken into account. Of the total bacterial species revealed by 16S amplicon profiling with relative abundance >10%, 38.7% of bacterial species were not revealed by bacterial culture; most of the time, the offending species was a *Corynebacterium*. This study showed that a careful interpretation of the result of the bacterial culture should be performed, and always with the support of cytology. To assess the overall bacterial population in suppurative otitis, the 16S amplicon profiling method seems to be a more accurate method but does not provide information on bacterial susceptibility.

**Abstract:**

A discrepancy between cytology and bacterial culture methods is sometimes observed in canine otitis externa. The objective of this study was to compare results from cytology, bacterial culture and 16S amplicon profiling. Twenty samples from 16 dogs with chronic suppurative otitis externa were collected. A direct cytological evaluation was carried out during the consultations. Aerobic bacterial culture and susceptibility were performed by an external laboratory used in routine practice. For 16S amplicon profiling, DNA was extracted and the hypervariable segment V1–V3 of the 16S rDNA was amplified and then sequenced with a MiSeq Illumina sequence carried out by the Mothur software using the SILVA database. A good correlation between cytology and bacterial culture was observed in 60% of the samples. Some bacterial species revealed by bacterial culture were present with low relative abundance (<10%) in 16S amplicon profiling. Some bacterial species revealed by the 16S amplicon profiling analysis were not identified with culture; most of the time, the offending species was a *Corynebacterium*. To conclude, a careful interpretation of the results of bacterial culture should be made and always be in agreement with the cytology. The 16S amplicon profiling method appears to be a more sensitive method for detecting strains present in suppurative otitis but does not provide information on bacterial susceptibility.

## 1. Introduction

Otitis externa, a common complaint in canine veterinary medicine, can be observed following predisposing, primary, secondary or perpetuating factors [1,2]. Allergic dermatitis is the most common aetiology of otitis externa (OE), accounting for more than 60% of the underlying primary causes [3]. The main pathogens reported in the literature are *Malassezia pachydermatis* and bacteria, with *Staphylococcus pseudintermedius* representing more than 70% of bacteria when cultured [4]. Suppurative otitis is usually observed as a chronic evolution of persistent or recurrent otitis, which can evolve for months or even years. This type of otitis is characterized by the clinical appearance of a whitish/slimy ear exudate. Classical diagnostic tools are cytology and bacterial culture. The main bacteria recognized in canine OE are *S. pseudintermedius, Enterococcus* spp., *Pseudomonas aeruginosa, Streptococcus* spp., *Corynebacterium* spp. and *Escherichia coli* with bacterial culture [5,6,7,8]. Colonization by opportunistic bacteria can thus represent a therapeutical challenge for the veterinarian. In the absence of first-line therapy response or the presence of bacilli, bacterial culture with evaluation of antibiotic susceptibility is recommended [9]. Nevertheless, in practice, a discrepancy between direct cytology and bacterial culture is sometimes observed. In some cases, the preconized antibiotic based on the bacterial susceptibility is ineffective (or the opposite). A previous study reported that cytology has better sensitivity than bacterial culture in canine otitis externa with a possible lack of concordance between the two methods [4]. This discrepancy could be explained by the presence of uncultured bacteria or the emergence of the fastest-growing bacteria. Recent studies using 16S amplicon profiling have shown that the bacterial population observed in canine otitis externa is more complex than expected based on the bacterial culture technique [10,11,12,13,14,15].

The objective of this study was to compare the data provided by direct cytology, bacterial culture and 16S amplicon profiling in dogs with chronic suppurative otitis externa.

## 2. Materials and Methods

### 2.1. Study Population and Clinical Assessment of Ear Canal

Sixteen dogs with chronic suppurative otitis externa were selected in our referral practice. The selection criteria were the history and the presence of clinical signs of suppurative otitis externa: ear pruritus and/or pain, head shaking and suppurative discharge from the ear. Due to the evolution time of otitis in our case series, many had been treated with topical and/or systemic treatments and were accepted for inclusion. For dogs treated before inclusion, a minimum of two weeks of treatment without improvement had been deemed ineffective, justifying a consultation in a referral clinic. Verbal consent was obtained from owners prior to sampling. The ethics review was not requested by the parent institution, because it was considered as a clinical observation study.

Three samples from one ear canal were performed for each dog, except for dog number 11 where both sides were sampled independently.

Some dogs were swabbed twice, weeks or months apart: for dog number 2, a second sample was taken 5 months after the first one, for dog number 4, a second sample was taken after 3 months, and for dog number 7, two samples were taken 1 month apart.

Clinical assessment of the ear canal using a modified OTIS3 score was performed [16]. The modified OTIS3 score assesses erythema (0–3), oedema/swelling (0–3), erosions/ulcerations (0–3) and exudate (0–3). Pain was assessed as present or absent but was not scored here. Only dogs with a modified OTIS3 score higher than 4/12 were enrolled. Clinical evaluation was performed by the same clinician for all dogs.

### 2.2. Cytological Evaluation

For cytological evaluation, the ear canals were sampled with a sterile swab by rubbing the skin surface between the vertical–horizontal junction for a period of 5 s. The swab sample was transferred to a slide, heat-fixed and stained with Diff Quick (RAL Diagnostics; Martillac, France). A semi-quantitative scale (from 1 to 4) to evaluate the types of organisms (bacilli, cocci and Malassezia) was used as reported by Budach and collaborators [17]. Inflammatory cells, when present on cytological evaluation, were not quantified but qualified as present or absent. Cytological evaluation was performed by the same clinician for all samples.

### 2.3. Aerobic Culture and Susceptibility Testing

For aerobic culture and antimicrobial susceptibility, the ear canal was sampled using eSwabs (Copan Diagnostics; Murrieta, CA, USA) by rubbing the skin between the vertical–horizontal junction for 5 s. The samples were sent to an external laboratory within 12 h. The samples were stored at 4 °C before analysis. Bacterial culture was initiated on blood agar plates TSS (Trypcase Soy agar + 5% sheep blood, BioMerieux Marcy l’Etoile, France), Columbia CAP selective agar (Becton Dickinson GmbH, Heidelberg, Germany) and MacConckey agar (Merck KGaA, Darmstadt, Germany). Microbial identification was performed with a matrix-assisted laser desorption ionization–time-of-flight mass spectrometry (MALDI-TOF MS) (Bruker Daltonics, Billerica, MA, USA). Antimicrobial susceptibility was assessed using the AST-GN97 panel (VITEK 2; BioMérieux, Marcy l’Etoile, France) for Gram-negative bacteria, or Kirby–Bauer disk diffusion for other bacteria. In the routine lab analyses, results were expressed as present without information on relative abundance.

### 2.4. Bacterial DNA Extraction and Amplicon Sequencing

The ear canal was sampled using FLOQSwabs (Copan Diagnostics, Murrieta, CA, USA) by rubbing the skin between the vertical–horizontal junction for 5 s and immediately stored at −18 °C until DNA extraction.

The DNA extraction, amplification and sequencing were performed as previously described [18,19].

Briefly, total bacterial DNA was extracted from the ear swabs with the DNEasy Blood and Tissue kit (QIAGEN Benelux BV; Antwerp, Belgium) following the manufacturer’s recommendations. Negative control samples with sterile swabs were included. This protocol was preceded by a bead-beating step with glass beads >106 μm and soda-lime glass beads (Sigma-Aldrich, Overijse, Belgium, Cat. G4649 and Z265926).

PCR amplification of the V1–V3 region of the 16S rRNA gene and library preparation were performed with the following primers (with Illumina overhand adapters): forward (5′-GAGAGTTTGATYMTGGCTCAG-3′) and reverse (5′-ACCGCGGCTGCTGGCAC-3′). Each PCR product was purified with the Agencourt AMPure XP beads kit (Beckman Coulter; Pasadena, CA, USA) and submitted to a second PCR round for indexing, using the Nextera XT index primers 1 and 2. After purification, PCR products were quantified using the Quant-IT PicoGreen (ThermoFisher Scientific; Waltham, MA, USA) and diluted to 10 ng/μL.

A final quantification, by quantitative (q)PCR, of each sample in the library was performed using the KAPA SYBR” FAST qPCR Kit (KapaBiosystems; Wilmington, MA, USA) before normalization, pooling and sequencing on a MiSeq sequencer using V3 reagents (Illumina; San Diego, CA, USA). Positive control using DNA from 20 defined bacterial species and a negative control (from the PCR step) were included in the sequencing run.

Raw amplicon sequencing libraries were submitted to the NCBI database under bio project number PRJNA844299.

### 2.5. Sequence Analysis and 16S rDNA Profiling

Sequence reads processing was as described previously using MOTHUR software package v1.421 (https://www.mothur.org, accessed on 1 March 2021) for alignment and clustering, and VSearch algorithm for chimera detection [20,21,22].

A clustering distance of 0.03 was used for OTU generation. 16S reference alignment and taxonomical assignment were based on the SILVA database (v1.38) (https://www.arb-silva.de, accessed on 1 March 2021) of full-length 16S rRNA sequences [23].

From 3,562,785 raw reads, 3,307,182 reads were obtained after cleaning (length and sequence quality) and chimera removal. Finally, 10,000 reads per sample were used for OTU clustering and taxonomic assignment. Good’s coverage estimator was used at the genus level as a measure of sampling effort for each sample, with a mean value of 99.75%. Negative controls, as a measure of determining erroneous results due to contamination, were not sequenced as there was no detectable amplification product in the samples. Suspected contaminants found in the controls (such as chloroplasts) were removed by filtering them from the OTU table, as previously described [24].

### 2.6. Correlation Analysis

To analyse the correlation between cytology/bacterial culture and between cytology/16S amplicon profiling (for 16S amplicon profiling only bacterial species with a relative abundance (RA) > 10% were used), a good correlation was considered if the morphological appearance of the bacterium was compatible (i.e., cocci observed at direct cytology with culture revealing *staphylococcus* spp.); a low correlation was considered if only one morphology of bacteria was observed on cytology but not on bacterial culture and vice versa (i.e., cocci AND bacilli observed on direct cytology with culture revealing only *staphylococcus* spp.); and a poor correlation was characterized by a discordance with the bacterial morphology between the two evaluation methods (i.e., bacilli observed on direct cytology with culture revealing only *staphylococcus* spp.).

For the correlation between bacterial culture and 16S amplicon profiling, the study was performed considering bacterial species with relative abundance >10% and with the closest matches to the 16S rRNA gene. A good correlation was considered if the same bacterial species were found in the bacterial culture and in the 16S amplicon profiling; a low correlation was considered if not all the bacterial species found in the bacterial culture were observed in the 16S amplicon profiling and vice versa; a poor correlation was noted if the bacterial species revealed by the bacterial culture were not found in the 16S amplicon profiling and vice versa.

## 3. Results

### 3.1. Population Description

A total of 16 dogs were enrolled with 20 ears sampled. The median age was 6 years (ranging from 1 to 13 years); the male-to-female ratio was 0.5. There were 12 different breeds with a high number of Cocker Spaniels (31%). Ten dogs (62.5%) presented unilateral otitis externa, and six dogs (37.5%) had bilateral otitis externa. Of the six dogs with bilateral otitis externa, four were Cocker Spaniels. The right-to-left ratio of the ear sampled in dogs with unilateral otitis externa was 4.5 (Table 1).

The evolution time of otitis was not always estimable because some owners reported an imprecise duration of months or years, but all dogs had a history of at least three months. None presented neurologic clinical signs. Of the 20 ears sampled, 13 samples were taken from treated ear canals (with an oral or topical antibiotic) and 7 from untreated ear canals.

### 3.2. Direct Cytology

Bacilli were observed in all the samples (100%), cocci in 16/20 (80%) and Malassezia in only 1/20 (5%). A mixed population of bacilli and cocci was observed in 16/20 (80%) samples. Neutrophils were present in 17/20 (85%) of the samples. The median cytological score was 3 for bacilli (ranging from 1 to 4), 1 for cocci (ranging from 0 to 4) and 0 for yeasts (ranging from 0 to 1) (Table 2).

### 3.3. Bacterial Culture

A total of 16 different bacterial species were isolated by aerobic culture. In the same ear sample, one bacterial species was isolated in 4/20 (20%), two bacterial species in 4/20 (20%), three bacterial species in 9/20 (45%) and four bacterial species in 3/20 (15%). The most frequently observed bacterium was *Pseudomonas aeruginosa* in 12/20 (60%), followed by *Staphylococcus pseudintermedius/delphini* in 10/20 (50%), *Proteus mirabilis* in 5/20 (25%) and *Streptococcus canis* in 5/20 (25%) (Table 2).

### 3.4. External Ear Canal 16S Amplicon Profiling

A total of 180 bacterial species from seven different phyla were isolated. At the phylum level, the Proteobacteria were the most abundant with a prevalence of 49.23% followed by Actinobacteria (30.11%), Firmicutes (13.33%) and Bacteriodetes (6.60%). The major genus was *Pseudomonas* with a prevalence of 43.33% (*Pseudomonas aeruginosa* 43.33%), followed by *Corynebacterium* with 28.25% (major species: *Corynebacterium auriscanis* 16.62% and *Corynebacterium jeikeium/amycolatum* 11.32%), *Porphyromonas* with 5.85% (major species: *Porphyromonas cangingivalis* 5.57%) and *Staphylococcus* (5.84%) (major species: *Staphylococcus delphini/pseudintermedius* 5.83%) (Table 3).

### 3.5. Correlation Analysis

#### 3.5.1. Correlation between Cytology and Bacterial Culture

A good correlation between cytology and bacterial culture was observed in 12/20 samples (60%) and a low correlation in 8/20 samples (40%). None showed a poor correlation (Table 2).

#### 3.5.2. Correlation between Cytology and 16S Amplicon Profiling

When bacterial species with relative abundance <10% were removed, good correlation between cytology and 16S amplicon profiling was observed in 7/20 samples (35%) and low correlation in 13/20 samples (65%). None showed a poor correlation.

#### 3.5.3. Correlation between Bacterial Culture and 16S Amplicon Profiling

When bacterial species with relative abundance <10% were removed, good correlation between bacterial culture and 16S amplicon profiling was observed in 2/20 (10%), low correlation in 13/20 (65%) and poor correlation in 5/20 (25%)

When the closest matches of the 16S rRNA gene were included, a good correlation was observed in 14/20 (70%) and a low correlation in 6/20 (30%), and none had a poor correlation.

In samples number 3, 4B, 11R, 12, 14 and 16, some bacterial species observed in bacterial culture were absent in 16S amplicon profiling (*Escherichia coli, Proteus mirabilis, Rhizobium radiobacter, Staphylococcus pseudintermedius, Staphylococcus schleiferi, Staphylococcus vitulinus* and *Staphylococcus epidermidis*).

Of the total of all bacterial species revealed by 16S amplicon profiling with relative abundance >10%, 38.7% of bacterial species were not revealed by the bacterial culture. Of these, 41.66% were aerobic species and 58.33% were anaerobic species. Most of the time, the offending species was a *Corynebacterium*. (samples number 3, 4B, 7B and 15). In three out of these four cases, samples were taken from ear canals treated with ear drops containing topical glucocorticoids and antibiotics (gentamicin, polymyxin or marbofloxacin). The other bacteria not identified by culture were *Porphyromonas cangingivalis* (samples number 11L and 13), *Fusobacterium* (sample 6) and *Bacteroides pyogenes* (sample 11L). For sample 14, 16S amplicon profiling provided an unusually high number of bacterial strains (anaerobic and aerobic bacteria) with the presence of *Bordetella petrii*, *Finegoldia magna*, *Clostridium* spp. and even *Pseudomonas aeruginosa*. The underlined bacterial species are obligate anaerobes and therefore cannot be revealed by aerobic culture.

## 4. Discussion

This study seems to confirm that Cocker Spaniels are very sensitive to otitis externa since they were the breed most affected in our series. The Cocker Spaniel and its crosses are well-known to have an increased risk of otitis [9,25,26]. In our series, the minimum complaint period was at least 3 months and, in some cases, years.

The present study also confirmed the difficulty in making a decision about the choice of antibiotic treatment due to the possible discrepancy between the different techniques used (cytology, bacterial culture or 16S amplicon profiling). This fact has already been reported in a study comparing cytology and bacterial culture in otitis externa [4]. As shown in Table 2, in some cases, the cocci or bacilli seen on cytology were not cultured. The susceptibility of these bacteria to antibiotics therefore cannot be assessed. Nevertheless, in chronic or recurrent otitis, bacterial culture with susceptibility testing remains the recognized method for identifying pathogens and guiding the choice of treatment [27,28].

In our study, when bacterial species with relative abundance <10% in 16S amplicon profiling (considered insignificant) were removed, a good correlation between bacterial culture and 16S amplicon profiling was observed in only 10% of samples. However, when all bacteria revealed in 16S amplicon profiling were included, a good correlation was observed in 70%. Bacterial culture could thus be considered a questionable method because, sometimes, bacterial species present at very low relative abundance in 16S amplicon profiling can grow in bacterial culture and may appear as dominant according to this technique. This observation is probably dependent on the bacterial strain.

*Pseudomonas aeruginosa* was the most prevalent bacterial species observed in bacterial culture and 16S amplicon profiling, with a presence in more than half of our cases. All cases with *P. aeruginosa* revealed in 16S amplicon profiling were also confirmed by a positive bacterial culture. This observation highlights that, at least for the detection of *P. aeruginosa*, bacterial culture appears to be a sensitive method. The presence of *P. aeruginosa* in canine otitis externa with/without otitis media has been reported in several studies [5,7,10,11,12,14,15,26,28,29,30,31,32,33]. A prevalence from 7.2% to 35.5% of *P. aeruginosa* was recorded in bacterial culture [5,7,26,28,30,31,33]. A higher prevalence of *P. aeruginosa* was noted in our study. The comparison between these data is difficult because the selection criteria are not the same (chronic suppurative otitis in our study). Comparing the relative abundance of *P. aeruginosa* in 16S amplicon profiling is much more difficult because not all the authors mentioned the precise numbers in their results [10,11,12]. In addition, a comparison between different techniques used for 16S amplicon profiling may induce bias [34].

Regarding the bacteria not cultured but revealed by the 16S amplicon profiling, *Corynebacterium* spp. and *Porphyromonas cangingivalis* were the most frequently observed. In general practice, *Corynebacterium* spp. are considered sensitive bacteria, and routine susceptibility testing is rarely performed [35]. A susceptibility test was requested for *Corynebacterium* spp. in some cases because they did not respond to the recommended treatment, and multidrug resistance was observed. Other studies have also described the existence of multidrug resistance in *Corynebacterium* spp. strains [36,37,38]. Henneveld and colleagues demonstrated that *Corynebacterium* spp. from canine otitis showed resistance to certain antibiotics commonly used in ear drops (11.11% for gentamicin, 36.36% for polymyxin B and 84.62% for marbofloxacin) [39]. According to our data, *Corynebacterium* spp. were the primary pathogen in 16S amplicon profiling found in five dogs (relative abundance ranged from 77.72% to 98.72%); one dog had a mixed bacterial population. With bacterial culture, only one dog had a bacterial population mainly composed of *Corynebacterium* spp., and three had a mixed flora. Moreover, four *Corynebacterium* spp. were not revealed by culture (found by 16S amplicon profiling with a relative abundance from 63.12% to 88.54%); these strains could have an unidentified antibiotic resistance. This observation is relevant for *Corynebacterium* spp. but could also be for other bacterial species.

Other uncultured bacteria found in 16S amplicon profiling in our study, mostly anaerobes, were also reported by Tang and colleagues: *Porphyromonas cangingivalis*, *Finegoldia magna*, *Bacteroides pyogenes* and *Fusobacterium* [12]. The question of the role played by these bacteria as opportunistic pathogens and the need to perform anaerobic bacterial cultures remain open. Omar and colleagues mentioned that biofilm formation could promote the growth of anaerobic bacterial species due to lack of oxygen [40]. In contrast, in some samples, bacteria were observed in culture while they were absent in the 16S amplicon profiling analysis. This observation draws our attention to possible contamination during the process. In other situations, some bacteria observed in culture were present with low relative abundance (<10%) in 16S amplicon profiling. A possible explanation could be the rapid growth of these bacterial species during the culture process.

By respecting the inclusion criteria, cases of suppurative otitis were selected. These criteria for suppurative otitis are based on the clinical appearance of a whitish/slimy ear exudate. This appearance of exudate is generally associated with the presence of pus. Surprisingly, 3 of the 20 samples contained no neutrophils. The whitish/slimy exudate could thus be due to the formation of a biofilm. Unfortunately, laboratory techniques confirming the biofilm formation were not used in our study. Bacilli were observed in all the samples (100%) and cocci in 16/20 (80%). Correlation with bacterial culture was good for 50% of the samples and low for the remaining 50%. The correlation with 16S amplicon profiling, when bacterial species with relative abundance <10% were removed, was good for 35% of the samples and low for 65%. None of the methods seem perfect. Concerning the bacterial culture, as we have shown previously, the growth of bacteria can depend on several factors. Nevertheless, for the 16S amplicon profiling, the discrepancy is surprising because, in essence, the DNA could not lie.

Direct cytology is recommended as a key step in the consultation for the assessment of otitis externa. This technique directly gives the clinician an idea of the bacteria (or yeast) involved in the process. A limitation in our study was the choice of routine staining for cytology (Diff Quick^®^). The use of Gram staining would have been more interesting to give a more precise idea of the bacterial population present, in particular for the presence of *Corynebacterium* spp., Gram-positive bacilli.

The major limitation of this study was the small number of samples and the absence of anaerobic culture. Another limitation was the bacterial culture analysis of the sample performed in dogs treated with antibiotics/steroids, which may interfere with bacterial growth. Even though these treatments failed to control the bacterial infection (the reason the cases were referred), the residual antibiotics may have interfered with the growth of some bacteria. For understandable reasons, it was of course impossible to stop the treatment(s) for a certain period of time before carrying out a bacterial culture.

## 5. Conclusions

In conclusion, this study demonstrated that no technique is perfect. As proven by other studies, to assess the overall bacterial population, the 16S amplicon profiling method seems to provide more accurate results but unfortunately does not give information regarding antibiotic susceptibility. The culture method provides results for bacterial species able to grow under specific conditions and is today the only way to guide the antimicrobial therapy. Nevertheless, the culture and susceptibility must sometimes be interpreted with caution because, as shown by cytology, some bacteria present seem not to be revealed by culture. On the other hand, *Pseudomonas aeruginosa*, a frequent pathogen in chronic suppurative otitis, is most often easily revealed by the culture. Considering that some bacteria cannot be cultured, their susceptibility to antibiotics cannot be assessed and could be the cause of treatment failure. The 16S amplicon profiling method could be useful in revealing those non-growing bacteria but is difficult to use routinely.

## Figures and Tables

**Table 1 vetsci-09-00366-t001:** Signalment and clinical data of 20 dogs with suppurative otitis externa.

Sample Number	Breed	Sex	Age (Years)	Unilateral or Bilateral	Ear Sampled	OTIS 3 (Erytema, Oedema/Swelling, Erosions/Ulcers, Exudation)
1	Cocker Spaniel	F	6	Unilateral	Left	6/12
2A	Golden Retriever	F	13	Unilateral	Right	4/12
2B	Golden Retriever	F	13	Unilateral	Right	8/12
3	Cocker Spaniel	F	9	Bilateral	Right	4/12
4A	Basset Hound	M	11	Unilateral	Right	6/12
4B	Basset Hound	M	11	Unilateral	Right	7/12
5	Cocker Spaniel	MN	11	Bilateral	Left	8/12
6	Cocker Spaniel	MN	11	Unilateral	Right	8/12
7A	Tibetan Terrier	FN	13	Unilateral	Left	5/12
7B	Tibetan Terrier	FN	13	Unilateral	Left	5/12
8	Rhodesian Ridgeback	FN	1	Unilateral	Right	7/12
9	German Shepherd Dog	FN	10	Bilateral	Right	11/12
10	Burmese Mountain Dog	FN	6	Unilateral	Right	9/12
11R	Munsterlander	M	8	Bilateral	Right	11/12
11L	Munsterlander	M	8	Bilateral	Left	12/12
12	German Shorthaired Pointer	MN	4	Unilateral	Right	9/12
13	Shiba Inu	F	6	Unilateral	Right	5/12
14	Hungarian Vizsla	M	2	Bilateral	Right	4/12
15	Cocker Spaniel	M	5	Bilateral	Left	11/12
16	Weimaraner	M	2	Unilateral	Right	4/12

MN neutered male; FN neutered female.

**Table 2 vetsci-09-00366-t002:** Comparison of cytology and bacterial culture (with estimated correlation). Bacteria in non-bolded type have a morphology of cocci and in bold type a morphology of bacilli. *Trueperella bernardiae* (underlined) is a Gram-positive bacterium with a morphology varying from bacillus to coccoid rods.

Sample Number	Cytological Index of Cocci	Cytological Index of Bacilli	Culture Results	Correlation
1	1	**3**	** *Pseudomonas aeruginosa* ** ** *Proteus mirabilis* ** *Enterococcus canintestini*	Good
2A	0	**3**	** *Pseudomonas aeruginosa* ** *Staphylococcus pseudintermedius*	Low
2B	1	**3**	** *Pseudomonas aeruginosa* ** *Streptococcus canis* *Staphylococcus pseudintermedius*	Good
3	1	**3**	** *Proteus mirabilis* ** *Staphylococcus schleiferi* * Trueperella bernardiae *	Good
4A	1	**3**	** *Corynebacterium auriscanis* **	Low
4B	1	**3**	*Staphylococcus pseudintermedius* *Staphylococcus delphini* ** *Escherichia coli* **	Good
5	1	**3**	** *Pseudomonas aeruginosa* ** ** *Escherichia coli* **	Low
6	0	**3**	** *Pseudomonas aeruginosa* ** *Streptococcus canis* * **Escherichia coli** * *Enterococcus faecalis*	Low
7A	2	**4**	*Streptococcus agalactiae* * **Pseudomonas aeruginosa** *	Good
7B	3	**4**	*Staphylococcus pseudintermedius* *Streptococcus canis* * **Pseudomonas aeruginosa** *	Good
8	3	**2**	*Staphylococcus pseudintermedius*	Low
9	1	**3**	** *Corynebacterium auriscanis* ** *Staphylococcus pseudintermedius* *Streptococcus canis*	Good
10	1	**3**	** *Pseudomonas aeruginosa* ** ** *Citrobacter koseri* ** *Enterococcus faecalis*	Good
11 R	2	**4**	** *Proteus mirabilis* ** ** *Pseudomonas aeruginosa* ** *Staphylococcus vitulinus* ** *Rhizobium radiobacter* **	Good
11 L	3	**4**	** *Proteus mirabilis* ** ** *Escherichia coli* **	Mild
12	3	**3**	** *Pseudomonas aeruginosa* ** *Streptococcus canis* *Staphylococcus epidermidis*	Good
13	3	**1**	*Staphylococcus pseudintermedius*	Low
14	4	**3**	** *Proteus mirabilis* ** ** *Citrobacter koseri* ** *Enterococcus faecalis* *Staphylococcus pseudintermedius*	Good
15	0	**4**	** *Pseudomonas aeruginosa* **	Good
16	0	**4**	** *Pseudomonas aeruginosa* ** *Staphylococcus pseudintermedius* *Staphylococcus epidermidis*	Low

**Table 3 vetsci-09-00366-t003:** Comparison of bacterial culture and 16S amplicon profiling results with their relative abundance (RA) of predominant OTUs. This table shows culture results and 16S rRNA for all samples. For each sample, the closest 16S rRNA matching the culture result and its relative abundance (RA) are described in % for each bacterium. Finally, all the bacterial species with a relative abundance above than 10% are represented in the last column in % for each sample. (Anaerobic species are given in red).

Sample Number	Culture Results	Closest 16S rRNA Match	RA (%)	16S rRNA > 10% RA	RA (%)
1	*Pseudomonas aeruginosa* *Proteus mirabilis* *Enterococcus canintestini*	*Pseudomonas aeruginosa* *Proteus mirabilis* *Enterococcus canintestini*	73.76% 24.57% 0.05%	*Pseudomonas aeruginosa* *Proteus mirabilis*	73.76% 24.57%
2A	*Pseudomonas aeruginosa* *Staphylococcus pseudintermedius*	*Pseudomonas aeruginosa* *Staphylococcus delphini/pseudintermedius*	99.01% 0.96%	*Pseudomonas aeruginosa*	99.01%
2B	*Pseudomonas aeruginosa* *Streptococcus canis* *Staphylococcus pseudintermedius*	*Pseudomonas aeruginosa* *Streptococcus canis* *Staphylococcus delphini/pseudintermedius*	83.76% 11.05% 1.80%	*Pseudomonas aeruginosa* *Streptococcus canis*	83.76% 11.05%
3	*Proteus mirabilis* *Staphylococcus schleiferi* *Trueperella bernardiae*	Not detected Not detected *Trueperella bernardiae*	ND ND 1.54%	*Corynebacterium jeikeium/amycolatum*	77.72%
4A	*Corynebacterium auriscanis*	*Corynebacterium auriscanis*	98.72%	*Corynebacterium auriscanis*	98.72%
4B	*Staphylococcus* *pseudintermedius*	*Staphylococcus* *delphini/pseudintermedius*	9.10%	*Corynebacterium auriscanis*	88.54%

	*Staphylococcus delphini*	*Staphylococcus* *delphini/pseudintermedius*	9.10%		

	*Escherichia coli*	Not detected	ND		
5	*Pseudomonas aeruginosa* *Escherichia coli*	*Pseudomonas aeruginosa* *Escherichia coli/Shigella*	97.16% 0.07%	*Pseudomonas aeruginosa*	97.16%
6	*Pseudomonas aeruginosa* *Streptococcus canis* *Escherichia coli* *Enterococcus faecalis*	*Pseudomonas aeruginosa* *Streptococcus canis* *Escherichia coli/Shigella* *Enterococcus faecalis*	62.54% 13.66% 2.18% 0.15%	*Pseudomonas aeruginosa* *Streptococcus canis* *Fusobacterium*	62.54% 13.66% 10.80%
7A	*Streptococcus agalactiae* *Pseudomonas aeruginosa*	*Streptococcus agalactiae* *Pseudomonas aeruginosa*	6.18% 89.7%	*Pseudomonas aeruginosa*	89.7%
7B	*Staphylococcus* *pseudintermedius*	*Staphylococcus* *delphini/pseudintermedius*	31.08%	*Corynebacterium auriscanis* *Staphylococcus* *delphini/pseudintermedius*	63.12% 31.08%
	*Streptococcus canis*	*Streptococcus canis*	4.67%		
	*Pseudomonas aeruginosa*	*Pseudomonas aeruginosa*	0.67%		
8	*Staphylococcus pseudintermedius*	*Staphylococcus delphini/pseudintermedius*	79.09%	*Staphylococcus delphini/pseudintermedius*	79.09%
9	*Corynebacterium auriscanis* *Staphylococcus pseudintermedius* *Streptococcus canis*	*Corynebacterium auriscanis* *Staphylococcus delphini/pseudintermedius* *Streptococcus canis*	97.50% 0.27% 0.02%	*Corynebacterium auriscanis*	97.50%
10	*Pseudomonas aeruginosa* *Citrobacter koseri* *Enterococcus faecalis*	*Pseudomonas aeruginosa* *Citrobacter koseri* *Enterococcus faecalis*	90.10% 7.73% 0.04%	*Pseudomonas aeruginosa*	90.10%
11 R	*Proteus mirabilis* *Pseudomonas aeruginosa* *Staphylococcus vitulinus* *Rhizobium radiobacter*	Not detected *Pseudomonas aeruginosa* Not detected Not detected	ND 99.98% ND ND	*Pseudomonas aeruginosa*	99.98%
11 L	*Proteus mirabilis* *Escherichia coli*	*Proteus mirabilis* *Escherichia coli/Shigella*	24.94% 0.76%	*Porphyromonas cangingivalis* *Proteus mirabilis* *Bacteroides pyogenes*	27.92% 24.94% 10%
12	*Pseudomonas aeruginosa* *Streptococcus canis* *Staphylococcus epidermidis*	*Pseudomonas aeruginosa* *Streptococcus canis* Not detected	88.97% 10.80% ND	*Pseudomonas aeruginosa* *Streptococcus canis*	88.97% 10.80%
13	*Staphylococcus pseudintermedius*	*Staphylococcus pseudintermedius*	0.08%	*Porphyromonas cangingivalis*	84.04%
14	*Proteus mirabilis* *Citrobacter koseri* *Enterococcus faecalis* *Staphylococcus pseudintermedius*	*Proteus mirabilis* *Citrobacter koseri* *Enterococcus faecalis* Not detected	9.85% 0.86% 0.79% ND	*Bordetella petrii* *Finegoldia magna* *Pseudomonas aeruginosa* *Clostridium*	38.73% 19.43% 14.99% 14.66%
15	*Pseudomonas aeruginosa*	*Pseudomonas aeruginosa*	0.02%	*Corynebacterium jeikeium/amycolatum*	81.58%
16	*Pseudomonas aeruginosa*	*Pseudomonas aeruginosa*	99.94%	*Pseudomonas aeruginosa*	99.94%
	*Staphylococcus* *pseudintermedius*	Not detected	ND		

	*Staphylococcus epidermidis*	Not detected	ND		

## Data Availability

Not applicable.

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
