# Peer review of "External Ear Canal Evaluation in Dogs with Chronic Suppurative Otitis Externa: Comparison of Direct Cytology, Bacterial Culture and 16S Amplicon Profiling"

_vetsci, 2022, doi:10.3390/vetsci9070366_

Round 1
Reviewer 1 Report
Abstract
Line 16 ...compare results from (delete coming)
Line 19 ..was evaluated by an external laboratory.
Introduction
Line 19. Bacterial culture and aerobic sensitivity testing was performed by an external laboratory
Line 36. Otitis externa is a common complaint in canine veterinary medicine and occurs as a consequence of predisposing, primary, secondary or perpetuating factors. Suggest cite original reference August JR. Otitis externa: A disease of multifactorial etiology. Vet Clins North Am Small Animal 1988, 18: 731-742.
With reference to "is a common complaint", you could cite: O'Neill DG, Volk AV, Soares T, Church DB, Brodbelt DC, Pegram C. Frequency and predisposing factors for canine otitis externa in the UK - a primary veterinary care epidemiological view. Canine Med Genet. 2021 Sep 7;8(1):7. They found 7.3% consultations
Line 39. You need to define what you mean by suppurative otitis externa
Line 40. growth, cytological examination of smears from the external ear canal can help....
Line 41. In the absence of first-line
Line 43. ...is sometimes noted. In some cases...
Suggest cite, for example, anomalies such as Dickson DB Love DN Bacteriology of the horizontal ear canal of dogs. JASP 1983 24: 413-421. A moderate proportion of "no growth detected" might be be explained by 16S?
Materials and Methods
Line 49. shown that the bacterial population
Line 59. Inclusion criteria were cases were referred for assessment of chronic otitis externa and many had been prescribed systemic or topical . medications
Line 60. These treatments had been deemed
Line 64. ...each dog, except for dog number 11..
Line 77. ....horizontal junction for a period of 5 seconds.
Line 78 ...was transferred to a glass, heat-fixed and stained with Diff Quick
Line 80 bacilli rather than rod (ref Choi N et al Vet Derm 2018, 29: 413-
Line 85. ..susceptibility the ear canal was sampled
Line 87 Samples collected were sent to an external
Line 100. .....horizontal junction for 5 seconds What then? any transport medium?
Line 120. A bit more detail of positive controls perhaps?
Line 144. low or poor, not mild
Line 146. Bacilli, not rods
Line 148. ditto
Line 153. ....profiling: a poor or low (not mild)
Line 162 Cocker spaniel please and in Table 1. And although 31% is high surely it needs to be compared to a reference population to be able to say "over represented" especially when only 16 dogs are in the study.
Table 1 . Cocker spanial, Rhodesian ridgeback, German shepherd dog, Hungarian Vizsla
Line 166. The definition of chronic otitis is hard to define. You could cite
Bajwa J. Canine otitis externa - Treatment and complications. Can Vet J. 2019 Jan;60(1):97-99. P
who stated "Otitis externa may be acute or chronic (persistent or recurrent) otitis lasting for 3 months or longer)"
Line 181 Table 2 Legend. Very high numbers (not massive) and either malassezial yeast or presumed M. pachydermatitis
Line 208. Low or poor correlation, not mild
Line 207 ditto
Line 209 ditto
Line 225. In contrast, when bacterial ...
Line 227. In three out of these four cases..
Line 228. ..topical antibiotics and presumably glucocorticoids? Is there any indication as to how recent the otic treatments were applied. It might be worth introducing another column in Table 1 to include this data.
Line 231. Was sample 14 analysed differently? You state "provided a more extended result", was this mentioned in the methods? Is this the only reason you found e.g. Finegoldia magna in just Dog 14? This needs to be mentioned in the discussion as Tang et al found it much more frequently in the ear canals of dogs with otitis externa
Discussion
General point. These dogs had previous treatments, although you don't mention how recent some of the treatment were? I think that the potential for previous use of glucocorticoids and antibacterial treatments to impact on your results needs a mention
Line 237 and 238 you can say Cocker spaniel a was most numerous but I don't think you can say is sensitive (i.e. predisposed) to suppurative otitis without comparative data. Again, it is Cocker spaniel
Line 239. If you describe what you mean by suppurative otitis externa in the introduction you don't need to describe it in the discussion
Line 241. ...the minimal period, not evolution
Line 243....difficulty in making a decision about the choice of
Line 248 recognised rather than preconized, I think
Lines 250-255 are bit difficult to understand and the use of the word questionable needs review
Line 259. carrier case?
Line 294. In contrast, not at the opposite
Line 296. Good point re contamination.
Line 302 and 303. in lint 176 you say neutrophils and here you say white blood. Re biofilm You did not describe you were looking for this in Methods and did not describe its presence or absence in Results. Suggest leave out
Line 304. cocci, not coccis
Line 305 low, not mild
Line 307 low not mild
Line 311. ...the assessment of otitis externa.
Line 316 bacillus not rod.
Author Response
Dear reviewers,
The authors would like to thank you for your constructive comments giving us the opportunity to improve our manuscript.
Find below our proposals:
Reviewer 1:
Abstract
- Line 16 ...compare results from (delete coming)
Modified
- Line 19 ..was evaluated by an external laboratory.
Modified
Introduction
- Line 19. Bacterial culture and aerobic sensitivity testing was performed by an external laboratory
Modified
- Line 36. Otitis externa is a common complaint in canine veterinary medicine and occurs as a consequence of predisposing, primary, secondary or perpetuating factors. Suggest cite original reference August JR. Otitis externa: A disease of multifactorial etiology. Vet Clins North Am Small Animal 1988, 18: 731-742.
Reference added
- With reference to "is a common complaint", you could cite: O'Neill DG, Volk AV, Soares T, Church DB, Brodbelt DC, Pegram C. Frequency and predisposing factors for canine otitis externa in the UK - a primary veterinary care epidemiological view. Canine Med Genet. 2021 Sep 7;8(1):7. They found 7.3% consultations
Reference added
- Line 39. You need to define what you mean by suppurative otitis externa
For better understanding we moved and completed a sentence from the discussion to the introduction:
Line 37: “Suppurative otitis is usually observed as a chronic evolution of persistent or recurrent otitis, which can evolve for months or even years. This type of otitis is based on the clinical appearance of a whitish/slimy ear exudate.”
- Line 40. growth, cytological examination of smears from the external ear canal can help....
Modified
- Line 41. In the absence of first-line
Modified
- Line 43. ...is sometimes noted. In some cases...
Modified
Suggest cite, for example, anomalies such as Dickson DB Love DN Bacteriology of the horizontal ear canal of dogs. JASP 1983 24: 413-421. A moderate proportion of "no growth detected" might be explained by 16S?
We are reluctant to add a refence for a paper published 40 years ago. This paper does not add interesting information.
Materials and Methods
- Line 49. shown that the bacterial population
Modified
- Line 59. Inclusion criteria were cases were referred for assessment of chronic otitis externa and many had been prescribed systemic or topical medications
Modified as: “Due to the evolution time of otitis in our case series, many had been treated with topical and/or systemic treatments and were accepted for inclusion.”
- Line 60. These treatments had been deemed
Modified
- Line 64. ...each dog, except for dog number 11..
Modified
- Line 77. ....horizontal junction for a period of 5 seconds.
Modified
- Line 78 ...was transferred to a glass, heat-fixed and stained with Diff Quick
Modified
- Line 80 bacilli rather than rod (ref Choi N et al Vet Derm 2018, 29: 413-
Modified
- Line 85. ..susceptibility the ear canal was sampled
Modified
- Line 87 Samples collected were sent to an external
Modified
- Line 100. …..horizontal junction for 5 seconds What then? Any transport medium?
The eSwabs from Copan Diagnostics contains liquid amies transport medium.
- Line 120. A bit more detail of positive controls perhaps?
We don’t agree. We explain that the normal procedure for that technique respects the classical protocol: Positive control using DNA from 20 defined bacterial species and a negative control (from the PCR step) were included in the sequencing run.
- Line 144. low or poor, not mild
Modified with “low”
- Line 146. Bacilli, not rods
Modified
- Line 148. Ditto
Modified
- Line 153. ....profiling: a poor or low (not mild)
Modified
- Line 162 Cocker spaniel please and in Table 1. And although 31% is high surely it needs to be compared to a reference population to be able to say "over represented" especially when only 16 dogs are in the study.
Modified with “high number” instead of “over representation”
- Table 1 . Cocker spanial, Rhodesian ridgeback, German shepherd dog, Hungarian Vizsla
Modified
- Line 166. The definition of chronic otitis is hard to define. You could cite
Bajwa J. Canine otitis externa - Treatment and complications. Can Vet J. 2019 Jan;60(1):97-99. P who stated "Otitis externa may be acute or chronic (persistent or recurrent) otitis lasting for 3 months or longer)"
Modified, we agree that it is not the chronicity what we mean but the time of evolution of the otitis. The stage of chronicity is a difficult topic to assess.
“The evolution time of otitis was not always estimable because some owners reported an imprecise duration of months or years, but all dogs had a history of at least three months. None presented neurologic clinical signs. Of the 20 ears sampled, 13 samples were taken from treated ear canals (with an oral or topical antibiotic) and 7 from untreated ear canals.”
- Line 181 Table 2 Legend. Very high numbers (not massive) and either malassezial yeast or presumed pachydermatitis
Modified, we agree it could be impossible to determine if it is really a Malassezia or another type of yeast. We prefer to simplify as: “yeasts”
- Line 208. Low or poor correlation, not mild
Modified
- Line 207 ditto
Modified
- Line 209 ditto
Modified
- Line 225. In contrast, when bacterial ...
Modified
- Line 227. In three out of these four cases..
Modified
- Line 228. ..topical antibiotics and presumably glucocorticoids? Is there any indication as to how recent the otic treatments were applied. It might be worth introducing another column in Table 1 to include this data.
Modified line 279 as: “In three out of these four cases, samples were taken from ear canals treated with ear drops containing topical glucocorticoids and antibiotics (gentamicin, polymyxin or marbofloxacin). »
Comment: we think the input of data in the table will decrease the readability of the table without giving relevant information.
- Line 231. Was sample 14 analysed differently? You state "provided a more extended result", was this mentioned in the methods? Is this the only reason you found e.g.Finegoldia magna in just Dog 14? This needs to be mentioned in the discussion as Tang et al found it much more frequently in the ear canals of dogs with otitis externa
Comment: Sample 14 was not analyzed differently. By more extended result we would highlight that we found several anaerobic species. I think our sentence was not correct.
Modified as: “For the sample 14, 16S amplicon profiling provided an unusual high number of bacterial strains (anaerobic and aerobic bacteria) with the presence of Bordetella petrii, Finegoldia magna, Clostridium spp. and even Pseudomonas aeruginosa..”
Discussion
- General point. These dogs had previous treatments, although you don't mention how recent some of the treatment were? I think that the potential for previous use of glucocorticoids and antibacterial treatments to impact on your results needs a mention
We suggest to modify the sentence in the § major limitation in line 427:
“Another limitation is the bacterial culture analysis of the sample performed in dogs treated with antibiotics/steroids which may interfere with bacterial growth. Even though these treatments failed to control the bacterial infection (the reason the cases were referred), the residual antibiotics may have interfered with the growth of some bacteria. »
Comment: we rephrase the line 75 for a better comprehension.
“For dogs treated before inclusion, a minimum of two weeks of treatment without improvement had been deemed ineffective, justifying a consultation in a referral clinic.”
- Line 237 and 238 you can say Cocker spaniel a was most numerous but I don't think you can say is sensitive (i.e. predisposed) to suppurative otitis without comparative data. Again, it is Cocker spaniel
Modified : “ This study seems to confirm that Cocker spaniels are very sensitive to otitis externa since they are the most numerous breed affected in our series.”
- Line 239. If you describe what you mean by suppurative otitis externa in the introduction you don't need to describe it in the discussion
Modified and placed in the introduction line 37
- Line 241. ...the minimal period, not evolution
Modified
- Line 243....difficulty in making a decision about the choice of
Modified
- Line 248 recognised rather than preconized, I think
Modified
- Lines 250-255 are bit difficult to understand and the use of the word questionable needs review
Proposal change line 302 “In our study, when bacterial species with relative abundance <10% in 16S amplicon profiling (considered insignificant) were removed, a good correlation between bacterial culture and 16S amplicon profiling was observed in only 10% of samples.”
- Line 259. carrier case?
Modified as: “Pseudomonas aeruginosa was the most prevalent bacterial species observed in bacterial culture and 16S amplicon profiling, with a presence in more than half of our cases.”
- Line 294. In contrast, not at the opposite
Modified
- Line 296. Good point re contamination.
We are sorry but we don’t not understand what you would like to change
- Line 302 and 303. in lint 176 you say neutrophils and here you say white blood. Re biofilm You did not describe you were looking for this in Methods and did not describe its presence or absence in Results. Suggest leave out
Modified: always neutrophils
For the second part of the comment (if we well understand your comment ?).
A whitish “purulent” aspect of an ear exudate observed in clinic is not always due to an accumulation of neutrophils or pyocytes (as it was observed in 3 cases). This aspect of exudate seems to be a consequence of the biofilm.
Modified as: « By respecting the inclusion criteria, cases of suppurative otitis were selected. These criteria for suppurative otitis are based on the clinical appearance of a whitish/slimy ear exudate. This appearance of exudate is generally associated with the presence of pus. Surprisingly, three of the 20 samples contained no neutrophils. The whitish/slimy exudate could then be due to the formation of a biofilm. Unfortunately, laboratory techniques confirming the biofilm formation were not used in our study. »
- Line 304. cocci, not coccis
Modified
- Line 305 low, not mild
Modified
- Line 307 low not mild
Modified
- Line 311. ...the assessment of otitis externa.
Modified
- Line 316 bacillus not rod.
Modified

Reviewer 2 Report
General comments:
1. Authors have presented the study stating the "Discrepancy between cytology and bacterial culture methods is sometimes observed in canine otitis externa and could be correlated with therapeutic failure" which I identified as the knowledge gap. I believe the research question and answers given have not filled the gap as 16 sequencing is not practical in clinical diagnosis.
2. Manuscript needs English and formatting check.
Method and materials:
3. Methods: In my opinion, inclusion of at least few healthy dogs as negative controls would have improved the study.
4. Methods: Inclusion of anaerobic culture would have needed for completion of the correlation results. If it was not included due to non-availability of facilities, it is best to acknowledge it as a limitation in this study when making conclusions.
5. Results: Because the study has included previously treated dogs, correlation between 16S with cytology or bacterial culture can be misleading due to capability of detection of cell free DNA in 16S. Meaning, 16S still could detect genetic materials of dead bacteria after treatment.
Line 25: Provision of bacterial species that were not identified with culture as a percentage would give the reader more context than saying some species.
Line 27: Suggest to remove "spp".
Line 29-31: In my opinion this sentence is redundant.
Line 36-37: This sentence has a little introductory value. It needs to be supported with listing/giving examples for primary, secondary and perpetuating factors.
Line 42-44: Reference are needed for this statement.
Line 59: Authors have mentioned that previous topically and/or systemically treated dogs were included into the study due to the chronicity of the disease and considering the treatment failure aspects. However this could affect the microbiota diversity. Those effects further vary with the type of medication, route of administration, duration of the treatment and the time from the treatment to the commencement of the study. In addition, 16S like sensitive molecular detection have the capability of detecting cell free DNA, therefore left over nucleic materials can mislead the analysis. So, in my opinion, this inclusion criteria may have introduced considerable variability in the results.
Line 66-69: Have the dogs treated after 1st sampling? if yes, what is the treatment schedule? Why there is variation in second sampling time points which adds more variability to the results?
Line 111: Standard representation of a primer sequence is 5' and 3'.
Line 174 and rest of the manuscript: The word be cocci (it is the plural term of coccus).
Line 204: Check the font type
Line 237-238: Sample size is too small to make this type of statement.
Line 300-304: Gave any bio-film forming organisms have detected by any the detection methods in those samples?
Author Response
Dear reviewers,
The authors would like to thank you for your constructive comments giving us the opportunity to improve our manuscript.
Find below our proposals:
Reviewer 2:
General comments:
- Authors have presented the study stating the "Discrepancy between cytology and bacterial culture methods is sometimes observed in canine otitis externa and could be correlated with therapeutic failure" which I identified as the knowledge gap. I believe the research question and answers given have not filled the gap as 16 sequencing is not practical in clinical diagnosis.
Of course, the 16 S is not a routine technique available for diagnosis but with the use of this new technique it gives us a different perspective on the bacterial population. All the things we assumed before can be different.
Our proposal : we therefore suggest deleting the second part of the sentence and could be correlated with therapeutic failure. In fact, it is more a conclusion or a putative theory.
- Manuscript needs English and formatting check.
Modified
Method and materials:
- Methods: In my opinion, inclusion of at least few healthy dogs as negative controls would have improved the study.
We do not agree, what would be the aim for this group. Healthy ear canals cannot be considered as a negative control because the presence of bacteria in heathy ear canals is well known (with a very low abundance). It is then not a negative control. It is another group with a low abundance thus giving a probability of more discrepancies.
- Methods: Inclusion of anaerobic culture would have needed for completion of the correlation results. If it was not included due to non-availability of facilities, it is best to acknowledge it as a limitation in this study when making conclusions.
We mentioned it in the limitation at the end of the paper: absence of anaerobic culture …
We do not perform anaerobic cultures for two reasons:
- The ear canal is a site very exposed to aerobic condition
- Routine dermatology protocol recommends aerobic culture only. Maybe the recommendation should be changed but so far no evidence supports the theory.
- Results: Because the study has included previously treated dogs, correlation between 16S with cytology or bacterial culture can be misleading due to capability of detection of cell free DNA in 16S. Meaning, 16S still could detect genetic materials of dead bacteria after treatment.
This remark is often reported in studies using “16S” but following discussion with specialists in this technique, it appears that the detection of DNA from dead bacteria is very limited due to the very high rate of recycling of DNA by active (living) bacteria. We do not believe that this could induce a bias in the results.
- Line 25: Provision of bacterial species that were not identified with culture as a percentage would give the reader more context than saying some species.
We desagree with the Abstract (it is detail) Maybe the information can be added in the discussion section L271… we suggest adding it later in the article…
L275: “Of the total of all bacterial species revealed by 16S amplicon profiling with relative abundance >10%, 38.7% of bacterial species were not revealed by the bacterial culture. Of these, 41.66% were aerobic species and 58.33% were anaerobic species..”
- Line 27: Suggest to remove "spp".
Modified
- Line 29-31: In my opinion this sentence is redundant.
We do not agree: this is a conclusion that will be found in all papers using 16S. The method gives a more extensive (accurate) idea of the bacterial population but of course no information on the bacterial susceptibility.
- Line 36-37: This sentence has a little introductory value. It needs to be supported with listing/giving examples for primary, secondary and perpetuating factors.
We do not agree: the purpose of this study is to analyze different techniques but not to review the pathogenicity of otitis. References are present for readers wishing to find more information.
- Line 42-44: Reference are needed for this statement.
We propose to add this reference to this sentence:
Line 42 : “In the absence of first-line therapy response or presence of bacilli, a bacterial culture with evaluation of sensitivity to antibiotics is recommended (3).”
Reference: William H. Miller Jr, Griffin CE, Campbell KL. Muller and Kirk’s Small Animal Dermatology. In: 7th éd. St Louis: Elsevier Health Sciences; 2012. p. 741‑66.
- Line 59: Authors have mentioned that previous topically and/or systemically treated dogs were included into the study due to the chronicity of the disease and considering the treatment failure aspects. However this could affect the microbiota diversity. Those effects further vary with the type of medication, route of administration, duration of the treatment and the time from the treatment to the commencement of the study. In addition, 16S like sensitive molecular detection have the capability of detecting cell free DNA, therefore left over nucleic materials can mislead the analysis. So, in my opinion, this inclusion criteria may have introduced considerable variability in the results.
We agree that the treatments can modify the microbiota diversity but it is not the topic of the study. Concerning the “dead” DNA, please see the reply/comment giving before.
- Line 66-69: Have the dogs treated after 1st sampling? if yes, what is the treatment schedule? Why there is variation in second sampling time points which adds more variability to the results?
Yes, dogs were treated (based on the bacterial susceptibility) but did not respond to treatment or recurrence was observed. The second sample was not used in the study to compare the results between the samples but to assess the situation at a given time, to compare the techniques.
- Line 111: Standard representation of a primer sequence is 5' and 3'.
Modified
- Line 174 and rest of the manuscript: The word be cocci (it is the plural term of coccus).
Modified
- Line 204: Check the font type
We are sorry to not understand what you would like to change…
?... Anaerobic species are notified in red
- Line 237-238: Sample size is too small to make this type of statement.
Agree. Modified as: “This study seems to confirm that Cocker spaniels are very sensitive to otitis externa since they are the most numerous breed affected in our series.”
- Line 300-304: Gave any bio-film forming organisms have detected by any the detection methods in those samples?
Modified as: “By respecting the inclusion criteria, cases of suppurative otitis were selected. These criteria for suppurative otitis are based on the clinical appearance of a whitish/slimy ear exudate. This appearance of exudate is generally associated with the presence of pus. Surprisingly, three of the 20 samples contained no neutrophils. The whitish/slimy exudate could then be due to the formation of a biofilm. Unfortunately, laboratory techniques confirming the biofilm formation were not used in our study.»
Comment: the biofilm formation evaluation was not in the objectives of our study.
Round 2
Reviewer 2 Report
I do not believe that the authors have adequately addressed the issues raised to improve the quality of the manuscript. However the manuscript has improved in some aspect which is commendable.
Author Response
Dear reviewer,
We have tried to improve the introduction giving more data and references.
For a better understanding of our results, we have also modified and completed the Table 2.
We have modified the conclusion.
We are thanking you for your help and welcome any specific comments that could improve the paper.